# Hypertrophic Osteopathy Concurrent with an Aberrant Right Subclavian Artery in a Dog

**DOI:** 10.3390/vetsci11060263

**Published:** 2024-06-07

**Authors:** Young-Rok Kim, Jung-Hyun Kim

**Affiliations:** Department of Veterinary Internal Medicine, College of Veterinary Medicine, Konkuk University, Seoul 05029, Republic of Korea

**Keywords:** aberrant right subclavian artery, dog, hypertrophic osteopathy, vascular ring anomaly

## Abstract

**Simple Summary:**

Hypertrophic osteopathy (HO) is a rare bone disease characterized by abnormal new bone formation and soft-tissue swelling in the limbs. In veterinary medicine, HO is mainly linked to different underlying disorders, and its development mechanisms are unclear. In this case, report, the only remarkable finding was an aberrant right subclavian artery (ARSA), which is a rare form of vascular ring anomaly in dogs, identified by computed tomographic angiography. This anomalous vessel induced mild dilation of the esophagus above it. Although ARSAs may be a possible cause of HO, the owner declined surgical correction because of potential complications after surgery. After palliative treatment with nonsteroidal anti-inflammatory drugs (NSAIDs) for pain relief, intermittent lameness improved; however, no significant changes were observed in HO during the follow-up period.

**Abstract:**

A 13-year-old spayed female cocker spaniel was presented with a 2-month history of swelling in several digits and intermittent hindlimb lameness. Radiographs revealed marked soft-tissue swelling and periosteal new bone formation without cortical bone destruction, characteristic of hypertrophic osteopathy (HO), in the distal parts of all extremities except for the right forelimb. However, no notable findings were detected in thoracic radiographs. An ultrasonography indicated cranial bladder wall thickening, which resolved following antibiotic therapy. Computed tomographic angiography identified a potential underlying cause as an aberrant right subclavian artery (ARSA) originating from the aortic arch, compressing the esophagus and causing mild esophageal cranial dilation to the aberrant vessel. No other intrathoracic or neoplastic lesions were observed. Gastrointestinal symptoms, such as regurgitation, were absent. Although an ARSA was likely the cause of HO, surgical correction was declined by the owner. To the best of our knowledge, this is the first reported case of HO concurrent with ARSA in dogs.

## 1. Introduction

Hypertrophic osteopathy (HO) is rare disorder characterized by pathological periosteal new bone formation and soft-tissue swelling of the limbs [1,2]. In animals, HO is frequently secondary to an underlying condition and is often associated with neoplastic or infectious/inflammatory pulmonary diseases, including primary or metastatic lung tumors, lung abscesses, and pneumonia [1,2,3,4,5,6]. HO can be secondary to extrapulmonary causes such as esophageal diseases (congenital megaesophagus, spirocercal granuloma, and foreign bodies) [7,8,9] and cardiovascular diseases (bacterial endocarditis, right-to-left shunting patent ductus arteriosus, and aortic thrombosis) [10,11,12]. In rare cases, HO may be related to intra-abdominal neoplasms in the absence of pulmonary metastasis [2]. Although the pathogenesis is not well established, several hypotheses, including neural reflexes, peripheral hypoxemia, and vascular endothelial growth factor production from platelets, have been proposed as potential contributors to its development [1,2].

An aberrant right subclavian artery (ARSA), a vascular ring anomaly leading to esophageal constriction and dilation, is a rare developmental abnormality in dogs [13,14,15]. The right subclavian artery normally arises from the brachiocephalic trunk and supplies blood to the right thoracic limb, thorax, and neck [16]. In cases of ARSAs, this artery arises directly from the aortic arch caudal to the left subclavian artery and then courses from the same location as the left subclavian artery toward the right thoracic limb, passing the esophagus along its dorsal aspect [14,15]. Generally, an ARSA is characterized by the lack of early-onset clinical signs and mild esophageal dilation; it may present as an incidental finding without clinical relevance [15,17]. However, similar to other vascular ring anomalies, dogs with ARSAs often develop clinical signs, such as regurgitation [16,18,19].

To the best of our knowledge, HO concurrent with ARSAs in dogs has not been reported. This case report describes the association between an ARSA and HO in a dog, which was identified by computed tomography (CT) angiography.

## 2. Case Description

A 13-year-old spayed female cocker spaniel was referred to the Veterinary Teaching Hospital of Konkuk University with a 2-month history of intermittent lameness and digital swelling in all limbs, except for the right forelimb. On presentation, the dog was in good general condition. A physical examination revealed that the second or fifth digits of the affected limbs appeared firm, swollen, and erythematous but were non-painful on palpation (Figure 1). The dog exhibited bilateral ocular discharge and conjunctival hyperemia. Skin lesions including pruritus, erythema, and crusts were observed on the lips, ventral neck, paws, and base of the tail.

## 3. Diagnostic Assessment, Intervention, and Outcomes

The complete blood count was within normal limits. Serum biochemistry revealed hyperglobulinemia (5.1 g/dL; reference range, 2.5–4.5 g/dL), increased C-reactive protein levels (3.6 mg/dL; reference range, 0.1–1 mg/dL), elevated alkaline phosphatase levels (285 U/L; reference range, 23–212 U/L), and hypertriglyceridemia (119 mg/dL; reference range, 10–100 mg/dL). Impression smear cytology of the skin lesions revealed degenerative neutrophils with numerous cocci. Radiographs showed periosteal new bone formation limited to the phalanges, particularly evident on the abaxial aspect with a spiculated appearance, which affected the second or fifth digits of the left forelimb and both hindlimbs, accompanied by marked soft-tissue swelling around the proliferative lesions (Figure 2A,B). There was no evidence of cortical bone destruction or articular involvement. This finding represents a characteristic feature consistent with HO. However, no remarkable findings were observed on the thoracic radiography (Figure 2C). An abdominal ultrasonography was performed to identify the possible causes of the lesion, and thickening of the bladder wall was identified. Urinalysis revealed proteinuria and hematuria with positive urine culture results. An antinuclear antibody test was performed to rule out other inflammatory diseases, yielding a negative result.

After 6 weeks of antibiotic therapy, the urinary bladder wall and skin lesions markedly improved (Figure 3). Furthermore, biochemistry profiles including globulin (4.3 g/dL) and C-reactive protein (0.4 mg/dL) levels returned to normal. Due to persistent hematological abnormalities (i.e., elevated ALP, hypertriglyceridemia), thyroid function was assessed by measuring the serum thyroid hormone levels. The results revealed low serum free T4 levels (0.4 ng/dL; reference range, 0.6–3.7 ng/dL) and high serum TSH levels (1.59 ng/mL; reference range, 0.05–0.42 ng/mL), indicative of hypothyroidism. Consequently, levothyroxine (Synthroid; Bukwang Pharm., Seoul, Republic of Korea) was administered orally at a dosage of 0.01 mg/kg twice daily. One month after initiating treatment, alkaline phosphatase (181 U/L), free T4 (2.5 ng/dL), and TSH (0.28 ng/mL) levels were within the reference range.

However, there was no improvement in digit swelling and intermittent lameness. Therefore, a CT examination was performed under general anesthesia to identify lesions that might not have been detected in the previous examination. The contrast medium (Omnipaque 300; GE Healthcare, Oslo, Norway) was injected intravenously through the cephalic vein using a power injector (OptiVantage DH; Mallinckrodt, St. Louis, MO, USA). Contrast-enhanced CT revealed an ARSA originating from the aortic arch caudal to the left subclavian artery, rather than from the brachiocephalic trunk. The ARSA transversed the esophagus along its dorsal aspect. The esophagus was compressed, and the area cranial to the ARSA was mildly dilated (Figure 4). No abnormalities in other organs, such as the lungs, were detected on the CT. To further understand the anatomy of the ARSA, three-dimensional reconstructions of CT angiography images were generated using a volume-rendering method (Figure 5).

Further surgical management for the ARSA was recommended, which the owner declined. Instead, the dog was treated symptomatically with oral carprofen at a dosage of 4.4 mg/kg once daily (Rimadyl^®^; Zoetis, Parsippany, NJ, USA). Although the vascular deformity persisted and osseous changes progressed, lameness was not observed during the 6-month follow-up after administration of carprofen (Figure 6).

## 4. Discussion

In dogs, secondary HO is primarily associated with neoplastic or infectious pulmonary diseases, as reported previously [1,2]. HO is a paraneoplastic syndrome caused by tumors originating from visceral organs, such as the kidney or bladder [20,21,22]. The present case showed irregular bladder thickening on ultrasonography, with a positive urinary culture. Canine bladder tumors exhibit clinical signs similar to those in dogs with urinary tract infections and are often accompanied by a concurrent bacterial infection [23,24]. However, in the present case, improvement of the lesion was confirmed after antibiotic treatment, indicating cystitis rather than a bladder tumor. Other hematological abnormalities resolved during the treatment of the hormonal disease that was identified after antibiotic therapy (Appendix A). Therefore, further CT examination was required, which revealed an ARSA concurrent with esophageal dilation.

Some cases of esophageal-associated causes of HO have been reported previously [7,8,9]. However, cases of HO secondary to a vascular ring anomaly affecting the esophagus have not been reported. This report describes an ARSA, a rare vascular anomaly in veterinary medicine, that compressed the esophagus and likely contributed to HO in the dog.

The pathogenesis of HO remains elusive, but several hypotheses have been proposed to explain the cause of the periosteal bone proliferation [1,2]. According to the neurogenic hypothesis, the response of the afferent fibers of the vagus nerve distributed in the thorax to the neural reflex stimulates efferent pathways that increase blood flow to the distal limbs, subsequently causing overgrowth of the connective tissue and periosteum [21,25]. A previous report of a dog with HO secondary to megaesophagus suggested that vagal afferent stimulation initiates the neural reflex causing HO [9]. This hypothesis is supported by the fact that vagotomy, without eliminating the underlying causes, successfully resolves bone proliferation and clinical signs [26].

Anatomically, the right vagus nerve courses ventrally to the ipsilateral subclavian artery, and its branches innervate the esophagus and regulate peristalsis [27]. When an ARSA passes over the esophagus, it can lead to esophageal compression and dilation [14,18]. In response to the dilation of the esophagus, afferent fibers of the vagus nerve are stimulated through mechanoreceptors [28,29]. In the present case, chronic stimulation of the vagus nerve, induced by anomalous vessel-associated esophageal dilation, might have been the underlying cause of HO.

Dogs with ARSAs are often asymptomatic but can develop clinical signs, such as regurgitation and subsequent aspiration pneumonia, depending on the degree of esophageal compression [15,17]. A previous report on vascular ring anomalies demonstrated that malformation of the subclavian artery alone did not lead to severe esophageal constriction [19,30]. In the present case, the dilation of the esophagus was unremarkable, and the dog did not show typical clinical esophageal signs, such as regurgitation. However, the chronic pressure of the ARSA on the esophagus and its stimulation of the vagus nerve may suffice to induce HO.

The ideal treatment for HO depends on the underlying cause [1]. In previous studies, complete regression of HO was observed after the elimination of primary lesions, including tumor resection [31,32]. In the present case, ARSA ligation or vagotomy was considered to relieve HO. Although two cases describing the successful surgical management of ARSA through ligation of the anomalous vessel have been reported, ligation and transection of the right subclavian artery might result in failure of adequate circulation to the blood-supplying body parts (right thoracic limb, ventral thorax, and superficial neck) [16,18]. The owner declined surgical correction of the ARSA because of the absence of gastrointestinal symptoms and concerns about the high risk of intrathoracic surgery in an old dog. We prescribed a nonsteroidal anti-inflammatory drug suitable for palliative care to provide symptomatic pain relief.

In conclusion, we report the first case of HO concurrent with an ARSA in a dog, notably without regurgitation. Although the dog did not undergo surgical treatment for the ARSA, other possible causes of HO were ruled out, suggesting the possible involvement of an ARSA in the development of HO. The pathophysiology of HO in the present case remains unclear; however, chronic vagal stimulation related to esophageal dilation secondary to a vascular anomaly was considered as a possible factor. Therefore, if no other abnormalities are identified on conventional imaging modalities (that is, radiography and ultrasonography), CT angiography and 3D volume-rendering images are recommended, even in the absence of gastrointestinal symptoms.

## Figures and Tables

**Figure 1 vetsci-11-00263-f001:**
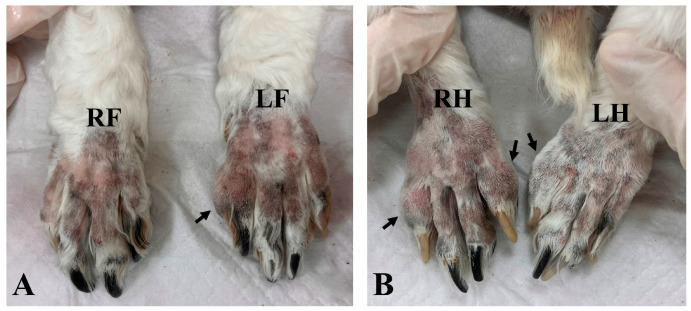
Distal forelimbs (**A**) and hindlimbs (**B**). Firm, non-painful swelling on the second or fifth digits of all limbs (black arrows), except for the right forelimb. RF, right forelimb; LF, left forelimb; RH, right hindlimb; LH, left hindlimb.

**Figure 2 vetsci-11-00263-f002:**
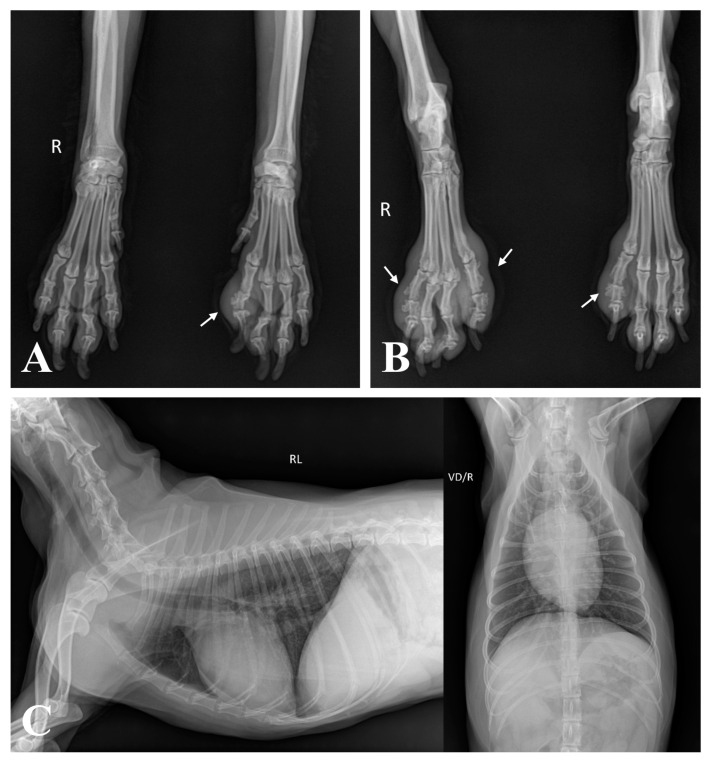
Radiographic images of all four limbs and thorax. (**A**,**B**) Periosteal new bone proliferation and soft-tissue swelling were evident in the second or fifth digit of the affected three limbs (white arrows) in the anterior–posterior view. (**C**) No remarkable findings were observed in the thorax on the right lateral and ventrodorsal view.

**Figure 3 vetsci-11-00263-f003:**
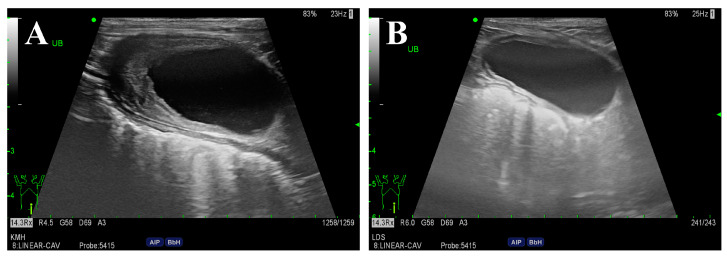
Ultrasonography of the urinary bladder before (**A**) and after (**B**) antibiotic therapy. The urinary bladder wall was initially thickened cranioventrally (5.2 mm). After 6 weeks of antibiotic therapy, significant improvement was observed, with the bladder wall returning to a normal thickness (1.9 mm).

**Figure 4 vetsci-11-00263-f004:**
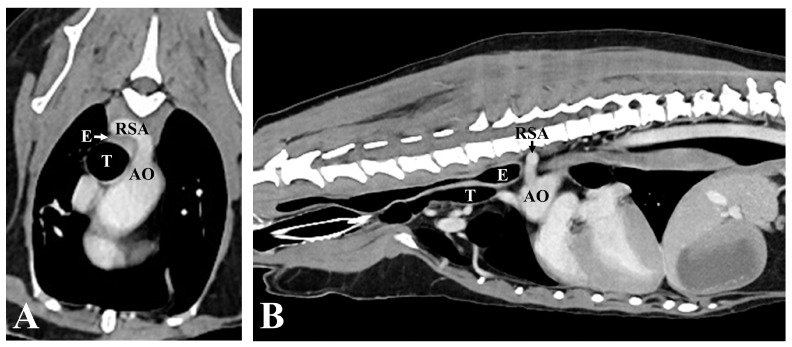
Computed tomographic angiography images. (**A**) Transverse image at the level of the third thoracic vertebra (T3). The esophagus is constricted between the RSA dorsally, and the trachea, ventrally. (**B**) Sagittal image of the thorax. Note that the esophagus located before the anomalous RSA is mildly dilated. AO, aorta; RSA, right subclavian artery; E, Esophagus; T, Trachea.

**Figure 5 vetsci-11-00263-f005:**
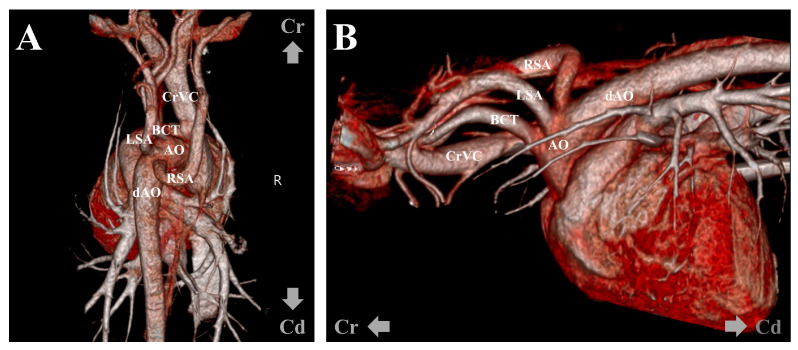
Volume-rendered three-dimensional reconstruction computed tomographic angiography images. Dorsoventral (**A**) and left-to-right projection views (**B**) show the right subclavian artery directly arising from the aortic arch caudal to the left subclavian artery. AA, aortic arch; dAO, descending aorta; RSA, right subclavian artery; LSA, left subclavian artery; BCT, brachiocephalic trunk; CrVC, cranial vena cava.

**Figure 6 vetsci-11-00263-f006:**
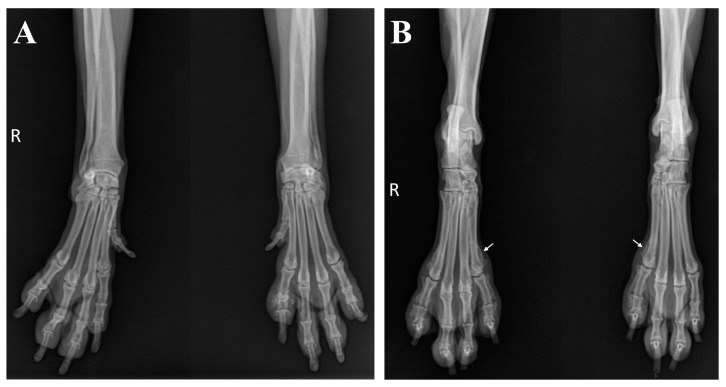
Radiographic changes in limbs during the follow-up period. No significant changes were detected in the forelimbs (**A**). However, periosteal new bone formation progressed proximally (white arrows) in the hind limbs (**B**).

## Data Availability

The data that support the findings of this study are available from the corresponding author upon reasonable request.

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
