# Peer review of "Hypertrophic Osteopathy Concurrent with an Aberrant Right Subclavian Artery in a Dog"

_vetsci, 2024, doi:10.3390/vetsci11060263_

Round 1

Reviewer 1 Report

Comments and Suggestions for Authors

Dear Authors:

Congratulations for your case report. In my opinion the manuscript was correctly structured and describes clearly the clinical case and the complementary tests. It would be nice to follow the case and obtain bone samples for analysis. Once again, congratulations for the work.

Author Response

We thank the reviewer for your thoughtful suggestions and insights. Please see the attachement for our reponse to comments. 

Reviewer 2 Report

Comments and Suggestions for Authors

I think it is an interesting case that is well structured, documented and presented. The introduction case presentation and discussion in general is adequate. It seems to me that the diagnosis of hypertrophic osteopathy is appropriate, however its association with the aberrant right subclavian artery (as cause - effect) lacks justification (as there was a clinical improvement without any change in the state of the mentioned blood circulation problem). Therefore, exploration of the possible link to cystitis that was treated and responded to treatment could also be a possibility.

After 6 months, was the improvement in clinical signs accompanied by improvements in radiographic changes in the limbs?

Author Response

(The authors gave the same response as above.)
